# COVID-19 Vaccination Acceptance in the Context of the Health Belief Model: Comparative Cross-Sectional Study in Punjab, Pakistan

**DOI:** 10.3390/ijerph191912892

**Published:** 2022-10-08

**Authors:** Rubeena Zakar, Ain ul Momina, Ruhma Shahzad, Sara Shahzad, Mahwish Hayee, Muhammad Zakria Zakar, Florian Fischer

**Affiliations:** 1Department of Public Health, University of the Punjab, Lahore 54590, Pakistan; 2Health Service Delivery in Punjab, King Edward Medical University, Lahore 54000, Pakistan; 3Department of Public Health and Primary Care, University of Cambridge, Cambridge CB2 1TN, UK; 4Oxford Policy Management, Islamabad 44000, Pakistan; 5Vice Chancellor Office, University of Poonch Rawalakot, Rawalakot 12350, Azad Jammu and Kashmir, Pakistan; 6Institute of Public Health, Charité–Universitätsmedizin Berlin, 10117 Berlin, Germany

**Keywords:** corona, SARS-CoV-2, vaccine, mass media, cues-to-action, health status

## Abstract

One of the models that could be used to understand the adoption of vaccine uptake is the Health Belief Model (HBM). The aim of this study is to assess the role of HBM constructs and Perceived Health Status (PHS) on the vaccination status of individuals and to understand the role of socio-demographic variables on HBM scoring. A comparative cross-sectional telephone survey was conducted among 1325 vaccinated (60.0%) and non-vaccinated (40.0%) individuals aged 40 years and above in July 2021 in Punjab province, Pakistan. A higher level of education was the strongest predictor of positive HBM. All constructs of HBM, PHS and cues-to-action were significant predictors of COVID-19 vaccination uptake, with perceived benefits as the strongest predictor. In order to expand the vaccination coverage, double-pronged interventions utilizing both information and communication technology and human resources should be designed that address each barrier perceived by individuals and understandably communicate the benefits of COVID-19 vaccination to the broader population.

## 1. Introduction

Following the rapid global spread of COVID-19, a public health emergency was declared worldwide in March 2020. Authorities at local, national and international levels introduced a range of precautionary and safety measures to reduce the spread of the virus. The rapid emergence of the COVID-19 pandemic created a situation in which health authorities went from recognizing a new pathogen (SARS-CoV-2) to developing and implementing an effective vaccine against COVID-19 in less than a year. Before, the mumps vaccine was reported to be developed and deployed in the shortest span, which took almost four years in the 1960s [1].

The United Kingdom was the first country to introduce, test and inoculate its population with the mRNA-based COVID-19 vaccine in December 2020 [2]. All countries followed starting their vaccination drive with different vaccines available in varying intervals. However, one of the biggest challenges that public health experts faced after the initiation of vaccination drive was COVID-19 vaccine hesitancy [3,4,5]. This has not only been observed in vulnerable populations [6], but also among health professionals and medical students [7]. The literature highlights various psychological [5,8], social [5,9], economic and personal factors associated with vaccine hesitancy worldwide [9].

Vaccine availability alone does not guarantee the effectiveness of vaccination drive [10]. Therefore, addressing vaccine hesitancy and fostering the confidence of masses in vaccine uptake is essential [8]. Widespread health education by awareness campaigns and strategies are needed to ensure the effectiveness and efficacy of the vaccination drive [3]. Researchers and public health experts focused on different epidemiological, behavioral and social models to understand the behavior regarding the uptake of the COVID-19 vaccine. One of the models that could be used to understand the adoption of healthy behavior is the Health Belief Model (HBM) [11].

HBM states that the perceived severity and susceptibility of a disease or a health risk as well as perceived benefits and barriers of a treatment or vaccine play a vital role in predicting the adoption of healthy and preventive behavior among individuals [12,13]. In the past, HBM has been conceptualized to understand the acceptance of masses towards the swine flu and seasonal influenza vaccine [14,15,16,17,18] as well as towards out-of-pocket payment for the hepatitis B vaccine [15]. Similarly, this model is now being used to understand people’s perceptions and acceptance towards the COVID-19 vaccine, because all theoretical constructs of HBM have a significant impact on the intentions of getting vaccinated [11,19]. In a survey experiment conducted by Zampetakis and Melas (2021) in Greece, 57% of the variance in the intention of getting the COVID-19 vaccine was explained by the theoretical constructs of HBM [19]. In addition to the role of the theoretical construct of HBM, self-reported health (SRH) also plays a significant role in predicting the health outcomes and adoption of preventive behavior among individuals [20]. SRH is an individual’s perceived social, mental and physical well-being [20], which is directly associated with various health outcomes and predicts the use of health services [20,21]. 

This paper is part of a broader post-vaccination survey of COVID-19. In the present study, SRH has been used to assess the Perceived Health Status (PHS). This study has been set up to introduce PHS as an additional construct of HBM in explaining COVID-19 vaccination acceptance. The specific objective of this study is to assess the role of HBM constructs including PHS on the vaccination status of individuals and to understand the role of socio-demographic factors (age, sex, region, income, and education) on HBM constructs. Lastly, this paper also evaluates the role of mass media as cues-to-action in health decisions in terms of vaccine uptake.

## 2. Health Belief Model

The framework of the HBM suggests that there are different negative and positive factors influencing the adoption of healthy behavior. The four main theoretical constructs of HBM include perceived susceptibility (P-SUS), perceived severity (P-SEV) of the health risk, perceived benefits (P-BEN) and perceived barriers (P-BAR) associated with the health behavior [11,14,15,16,19,22,23]. P-SUS defines the belief of people regarding their susceptibility or risk to catch the disease in question. People will adopt the healthy behavior if they believe that they are susceptible to a particular disease [14,15,16]. P-SEV is an individual’s belief regarding the severity or intensity of the disease. This component of HBM tells that if people believe in the severity of any specific disease, they are most likely to follow the precautionary measures to avoid the negative outcome of the disease [19]. While P-BEN and P-BAR are an individual’s belief regarding the benefits and constraints in taking up a preventive or healthy behavior, respectively. The people who have a strong belief that the target behavior, such as vaccination utilization, will have a positive benefit are more likely to get immunized. However, if they perceived strong barriers that prevent them from adopting the preventive behavior, then they are less likely to follow healthy behaviors [23]. Other factors that directly influence the health behavior decisions include socio-demographic factors of people and cues-to-action. The HBM is one of the widely accepted models to predict vaccine hesitancy among populations [24]. Past studies have revealed a significant role of theoretical constructs of HBM in the vaccination uptake of various communicable diseases, such as influenza [14,15,16,17,18,23,24], swine flu [14,15] and hepatitis B [15]. Similarly, recent studies have also highlighted the theoretical constructs of HBM as significant predictors of COVID-19 vaccination uptake [11,19,24]. 

One of the cross-sectional surveys conducted in China reveals P-SUS, P-BAR and P-BEN to be the significant predictors of COVID-19 vaccine hesitancy, with higher levels of P-BAR being the most significant predictor [25]. Likewise, another study conducted in Hungary found that P-BAR significantly reduces the likelihood of COVID-19 vaccination uptake, whereas P-BEN increases the probability of vaccination uptake by four folds [26]. On the other hand, other studies also highlight the role of P-SEV and P-SUS to be significant predictors of COVID-19 vaccination uptake [27,28]. Similarly, a study conducted in Pakistan showed that the fear of contracting COVID-19 (P-SUS) increases the likelihood of vaccine acceptance, while religious inhibitions (P-BAR) reduces the likelihood of vaccination uptake [29]. A systematic review also highlighted the theoretical constructs of HBM to significantly predict vaccination hesitancy, with P-BAR and P-BEN being the most commonly significant theoretical HBM constructs, where P-BAR and P-BEN are directly and inversely associated with vaccine hesitancy, respectively [30]. 

## 3. Materials and Methods

### 3.1. Study Design

A comparative cross-sectional study was conducted among 1325 vaccinated (60.0%) and non-vaccinated (40.0%) individuals aged 40 years and above from 8 July to 26 July 2021 in Punjab province, Pakistan. More than half of Pakistan’s population is residing in its largest province Punjab. The age group of 40 years and above was chosen because vaccine roll-out, starting from senior citizens, was followed in a decreasing age order. We assumed that at the time the present study was conducted, the majority of individuals in this age group had been vaccinated and only reluctant and/or hesitant individuals were non-vaccinated. Contact details of the target population were provided by Primary and Secondary Healthcare Departments in Punjab. Official landlines were used for data collection via a telephone-based survey. We estimated the sample size by using Cochran’s formula with the assumptions of 95% confidence interval, 3% margin of error, and adjustment for non-response. According to this, the total required sample was calculated to be 1284. Out of a random sample of 3550 individuals, which were reached via telephone, 1350 completed the questionnaire. We excluded 25 questionnaires because responses to more than 20% of questions were missing. This led to an overall response rate of 38%. 

Telephone interviews were chosen for data collection due to the mobility restriction imposed by the pandemic. We used a standardized questionnaire with close-ended questions. The questionnaire used was developed after extensive literature review and under the guidance of public health experts and bio-statisticians. A pre-test on 10% of the minimum required sample (128 respondents) was conducted. These data were not included in the final sample. Data were collected by a team of 15 persons who also assessed in the pre-test whether there were any redundant, repetitive, unclear or difficult-to-comprehend questions. After revising the questionnaire according to the comments and feedback provided in the pre-test, the final version of the questionnaire was developed.

### 3.2. Measures

Overall, the questionnaire had six sections. However, only four of the concerned sections are relevant to this analysis. The first section measured the socio-demographic profile that includes age, sex (male, female), religion (Islam, Christianity, other), educational level, monthly household income (≤PKR 20,000, PKR 20,001–50,000, PKR 50,001–100,000, above PKR 100,000), employment status, area of residence (rural, urban, semi-urban), city of residence, marital status, and number of children (if married). The second section focused on measuring media sources (television, mobile phone, newspaper, internet, and social media) used and most relied on for getting COVID-19-related information. This question has been used to assess cues-to-action. The third section measured the vaccination status and health status of the individuals. Respondents were asked whether they are vaccinated or not. This was followed by a question of SRH/PHS, allowing the respondents to rate their current health status using a 5-point Likert scale ranging from “very poor” to “very good” that was further recoded into three categories (poor, fair, good) for bivariate analyses. Usually, for measuring PHS, a person’s overall health status is measured based on their own subjective assessment [31]. 

The last section focused on measuring the perception of individuals regarding COVID-19 infection and vaccination according to the theoretical constructs of HBM. All the questions related to HBM were developed on the basis of an extensive review of literature on the topic. P-SUS is measured through one question (“Do you believe that there are confirmed or suspected cases of COVID-19 in the country?”) with three categories (yes, no, not sure). P-SEV was measured by the question “Do you believe that there is a threat of COVID-19 infection?” with a three-point scale (low, fair, high). For further analyses, the option “low” was recoded as “no” while the options “fair” and “high” were combined as “yes”. P-BEN were measured by two questions: “Do you perceive any benefits of getting vaccination?” and “Do you think that COVID-19 vaccination is an effective way of controlling and preventing the virus? ”Yes” to one or both questions was considered as a “Yes” in the final variable, while “No” to both questions was considered as a “No” in the final variable. P-BAR were also calculated by two questions: “Do you perceive any barriers in getting vaccination?” with “Yes” and “No” categories. If the respondents replied positively to this question then they were asked about the details of barriers through checklist including various barriers (distance to COVID-19 vaccine centers, cost of visit to obtain the vaccine, waiting time at the center, fear of getting infection at the center, lack of trust on the vaccine, lack of trust on the healthcare worker, perceived side effects of vaccine, false/misinformation about the vaccine and any other). Response of “yes” to any barrier was considered “Yes”, while responses of “No” for all barriers was considered “No” in the final variable. Figure 1 shows the pictorial representation of the variables used in the present study.

### 3.3. Statistical Analysis

For analyzing the role of socio-demographic factors on HBM, all theoretical constructs of HBM were computed. The role of having a high P-SUS, P-SEV and P-BEN is assumed to have a positive impact on adopting a preventive behavior, whereas the role of having a high P-BAR is considered to negatively impact healthy behaviors. For that reason, the direction of P-BAR was changed by transforming the variable and then all four theoretical constructs were computed. The value of the final variable ranged from 0 to 4, where 0–2 is considered as low scoring on HBM, which translates into “Poor COVID-19 knowledge”, while the range of 3–4 is considered as a high score on HBM, translating into “Good COVID-19 knowledge”.

For statistical analysis, SPSS version 26 (IBM Corp. Armonk, NY, USA) was used. Descriptive analysis is presented in percentages, frequencies or means, including standard deviation. For further analysis, Odds Ratios (ORs) and Adjusted Odd Ratios (AORs) were calculated by simple binary logistic regression and multivariate logistic regression, respectively. OR is calculated to measure the odds of vaccination uptake for each predicting variable individually, whereas AOR is calculated to measure the odds of vaccination uptake when all the predicting variables significant at *p* = 0.2 were adjusted collectively. Hierarchal logistic regression analysis was applied in eight steps to assess the predictive power of each variable in the overall model in COVID-19 vaccine uptake.

### 3.4. Ethical Considerations

The study protocol has been approved by the Institutional Ethics Review Board, University of the Punjab (D/No: 182/DFEMS/PU). All respondents were assured about the privacy, confidentiality and anonymity of the data and were told about their voluntary participation. Informed consent was taken from all participants.

## 4. Results

### 4.1. Characteristics of Study Participants

A total of 1325 respondents were included in analysis. A total of 60% of the total respondents were completely vaccinated, while the rest was not vaccinated. The majority of respondents were male (73.4%) and in the age group 40–49 years (48.8%), with a mean age of 51.2 years (SD ± 9.34). In addition to this, most of the respondents resided in urban areas (77.7%), follow Islam as a religion (96.1%), had a monthly income between PKR 20,001-50,000 (44.7%), were currently married (94.1%), had 3–4 children (48.0%), middle level of education (36.0%), high access to media (42.7%), and were employed (65.1%) (Table 1).

Overall, slightly more than half of the respondents (56.6%) had good COVID-19-related knowledge. The AOR indicate that being a resident of urban areas (AOR = 1.40, 95% CI: 1.02–1.92), having a middle (AOR = 1.82, 95% CI: 1.24–2.68) and high level of education (AOR = 2.36, 95% CI: 1.45–3.83), being currently married (AOR = 2.03, 95% CI: 1.02–4.03), and having 1–2 children (AOR = 0.03, 95% CI: 1.04–5.08) significantly predict having good COVID-19 knowledge (Table 2).

### 4.2. HBM and Vaccination Status

The majority of respondents reported perceived susceptibility (66.1%) and severity (70.8%) of COVID-19 infection, as well as perceived benefits (67.8%) and barriers (51.4%) of COVID-19 vaccination and good PHS (66.6%). All these factors were significantly associated in bivariate analyses and regression analyses with the vaccination status of respondents (*p* < 0.01). Data predict that people having high P-SUS (OR = 2.05, 95% CI: 1.62–2.58), high P-SEV (OR = 1.58, 95% CI: 1.25–2.01), high P-BEN (OR = 5.18, 95% CI: 4.04–6.65), low P-BAR (OR = 4.05, 95% CI: 3.20–5.14), and fair (OR = 2.56, 95% CI: 1.56–4.21) or good PHS (OR = 2.78, 95% CI: 1.73–4.45), and having high access to mass media (OR = 2.09, 95% CI: 1.11–3.92) are more likely to get vaccinated. The results portray P-BAR and P-BEN to be the strongest predictor in vaccination status (Table 3).

Table 4 describes the results of the multiple hierarchal regression analysis, which was performed on eleven variables (P-SUS, P-SEV, P-BEN, P-BAR, PHS, age, sex, region, monthly family income, access to mass media, and educational level) in eight steps. In the first four steps, all four constructs of HBM were added one by one to analyze the predictive value of each variable. P-SUS, P-SEV and P-BAR significantly predict the vaccination status of individuals (model 3). However, the addition of P-BEN in model 4 reduces the role of P-SEV and makes its role insignificant (AOR = 1.29, 95% CI: 0.97–1.70). 

Similarly, PHS significantly predicts the vaccination status (*p* < 0.01) in model 5, while P-SEV remains insignificant. In the next step, age, sex, region and income were added, collectively considering them as socio-demographic variables. 

It can be seen in model 6 that having high P-SUS (AOR = 1.68, 95% CI: 1.23–2.30) and P-BEN (AOR = 3.93, 95% CI: 2.88–5.34), low P-BAR (AOR = 3.35, 95% CI: 2.53–4.43), average (AOR = 2.88, 95% CI: 1.48–5.59) or good PHS (AOR = 2.41, 95% CI: 1.27–4.54), being in the age group of 60–69 years (AOR = 1.73, 95% CI: 1.13–2.65) and residing in urban areas (AOR = 2.34, 95% CI: 1.66–3.30) are factors associated with a higher likelihood to get vaccinated against COVID-19, while being female is associated with lower likelihood to get vaccinated (AOR = 0.53, 95% CI: 0.38–0.72) and the predictive role of monthly family income remains insignificant. 

In the seventh step, access to mass media was added. However, it does not make any significant difference except for making the predictive role of having a monthly family income of above PKR 100,000 (AOR = 0.50, 95% CI: 0.28–0.89) significant. 

In the last step, educational level has been added considering it as the strongest predictor of positive HBM (which is good COVID-19-related knowledge). It can be seen in model 8 that having a middle (AOR = 3.92, 95% CI: 2.50–6.16), secondary (AOR = 4.38, 95% CI: 2.44–7.84), and higher level of education (AOR = 4.53, 95% CI: 2.60–7.90) compared to no formal education significantly predicts COVID-19 vaccine uptake, while reducing the predictive value (AOR) of each significant predictor very slightly, except for the AOR of having average or good PHS, which increased. 

Overall, the data suggest that P-SUS, P-BEN, P-BAR, and average or good PHS significantly predict the COVID-19 vaccine uptake in respondents beyond the presence of socioeconomic variables, while the role of P-SEV and access to mass media remains insignificant with education level as the strongest predictor, followed by P-BEN, P-BAR and PHS, respectively.

## 5. Discussion

Various cross-sectional studies have evaluated the role of theoretical constructs of HBM in order to predict the adoption of healthy or preventive behaviors such as for influenza [14,15,16,17,18,32,33], swine flu [14,15] and hepatitis B [15] vaccine uptake. Considering the vaccination hesitancy related to COVID-19, epidemiologists and psychologists showed great interest in evaluating the role of theoretical constructs of HBM in the vaccination uptake [11] as well as in the adherence to precautionary measures [34] and preventive behavior of COVID-19 [35]. 

The results of this study reveal that despite the efforts that have been made at the local, national and international level for informing people about the risk of COVID-19 and vaccination benefits, a reasonable proportion of the adult population in our sample underestimated the risk of disease and perceived barriers in getting themselves vaccinated. However, this study provides a further piece of evidence that all theoretical constructs of HBM as well as PHS have direct and significant influence on the predicting intention of getting the COVID-19 vaccine, with P-BEN having the largest impact on COVID-19 vaccination acceptance followed by P-BAR, fair or good PHS, P-SEV and P-SUS, respectively. However, when adjusted for socio-demographic variables, the role of P-SEV becomes insignificant. Moreover, this study finds cues-to-action, such as access to mass media, to be positively associated with the vaccination status of individuals, but similarly, its role became insignificant when adjusted with other predictors.

Our results echo the findings of previous studies. For example, in a survey experiment study conducted in Greece [19], all theoretical constructs of HBM were found to be significantly associated with COVID-19 vaccination uptake, with P-BEN being the strongest predictor followed by P-BAR, P-SEV and P-SUS, respectively. Similarly, in a cross-sectional study conducted in Malaysia, Wong and colleagues [11] found higher levels of P-SEV, P-SUS and P-BEN to be positively associated with the intentions of getting vaccinated against COVID-19, while low levels of P-BAR were positively associated with COVID-19 vaccine uptake intentions. Shmueli [36] also found P-SEV, P-BEN and cues-to-action as significant predictors of COVID-19 vaccine uptake intention. However, no significant association of PHS with vaccine intention was found in both of these studies [11,36].

A study conducted in Hong Kong reported P-SEV, P-BEN, P-BAR, SRH (PHS) and cues-to-action to be significantly correlated with COVID-19 vaccine acceptance [37], yet Mercadante and Law [38] found P-BEN and cues-to-action to positively predict the intention to get vaccinated. However, in another study conducted on the intentions for the uptake of H1N1 influenza vaccine, no evidence was found for a significant relation for P-SUS and P-SEV, whereas P-BEN and P-BAR were significant predictors of vaccine uptake [32]. Likewise, in a study conducted in Pakistan, Shah and colleagues (2021) [39] evaluated the role of HBM constructs in the adherence of preventive measure of COVID-19 and found the significant role of P-BEN, while the role of all other constructs in the adoption of preventive behavior, including cues-to-action, remained insignificant.

Previous studies correlate socio-demographic variables with the intention and likelihood of getting vaccinated against COVID-19 and found that males are more likely than females to accept COVID-19 vaccination [11,36,40]. Literature also indicates that the higher immunization hesitancy among females is due to the belief that COVID-19 vaccines may cause infertility [41]. Our study results confirm the lower likelihood for vaccine uptake among women, even when adjusted with theoretical constructs of HBM and PHS. In addition to this, previous research has shown that age [36,38,40,42,43,44,45,46,47,48,49], income [11,38,45,48,50,51,52,53], education [11,36,43,45,46,47,50], working status [40,47,54,55], marital status [51,52,53], access to mass media, and area of residence [43,46,52,56,57,58] are significant predictors of COVID-19 vaccine acceptance. Our study provides insights that when HBM constructs and PHS are adjusted for socio-demographic variables, the role of gender, age, area of residence, income and education remain significant. Level of education was even the strongest predictor of COVID-19 vaccine uptake, while the role of access to mass media (cues-to-action) became insignificant. In addition to this, this study evaluates the role of socio-demographic variables in scoring well on HBM. It was found that individuals who have a middle, secondary, or higher level of education, reside in urban areas, have a monthly income of more than PKR 20,000, have high access to media, and are currently married are more likely to score well on HBM and have good COVID-19-related knowledge, while sex, age, number of children, and working status are not significant predictors of scoring well on HBM.

In summary, the findings of the present study shed light on the direct implications of the theoretical constructs of HBM and PHS, as well as on those people who are less likely to score positively on the HBM. These results help us to understand COVID-19 vaccine hesitancy among the broader population. The present study finds that all four theoretical constructs of HBM, PHS and cues-to-action significantly predict the COVID-19 vaccination intention. The findings reveal that higher educational level is the strongest predictor of COVID-19 vaccine uptake. These findings highlight the importance and need of education. Even though the influence of access to mass media became insignificant when adjusted with other predictors of vaccination uptake, the role of media health literacy campaigns regarding COVID-19 infection and vaccination cannot be neglected. This can be informed about in public service messages on television and social media and COVID-19-related messages as caller tunes, as far as the population is able to differentiate between evidence-based materials and fake or misleading information. 

However, the fact cannot be neglected that when educational campaigns are integrated with information and communication technology (ICT), it subjects the vulnerable and marginalized communities to a more vulnerable state [59,60,61], especially in the socio-demographic context of Pakistan, where more than half of the population resides in rural areas [62]. Therefore, there is a need to design double-pronged health education and information campaigns that integrate ICT as well as human resources. The present study recommends that in order to expand the vaccination coverage, interventions should be designed in a way that target the inclusion of marginalized communities as well as address the perceived barriers hampering the uptake of vaccination. Tailored strategies like door-to-door vaccination campaigns with health education messages integrating ICT as well as health professionals and healthcare workers could be utilized to specifically target the marginalized communities and refusal cases. Lastly, considering the significant estimates of PHS in predicting the COVID-19 vaccine uptake, it is recommended for future studies to further evaluate its impact as an additional construct of HBM with varying disease models.

## 6. Conclusions

This study shows the significant role of all the theoretical constructs of HBM and PHS in predicting preventive behavior for vaccination uptake during the COVID-19 pandemic. P-BAR and P-BEN have been the strongest predictors for healthy behavior and having a high level of education as the strongest predictor of positive scoring at HBM. Given that the high P-BAR is the strongest negative predictor of the vaccination status of respondents, there is a need to identify, address and eliminate barriers to vaccination, so more people could get the vaccination. Given the impact of P-BEN as a positive predictor of vaccine uptake, the right and accurate information needs to be disseminated in order to improve vaccination coverage. Health education and promotion strategies integrating both ICT and human resources could be designed to develop a more inclusive approach to address and eliminate P-BAR as well as to promote the benefits associated with vaccination uptake.

### 6.1. Future Research

Given the significant role of PHS, there is a need to further investigate its role as an additional construct of HBM in predicting the adoption of healthy or preventive behaviors among people of different socio-demographic profiles. Furthermore, there is a need to investigate this model on the general population of varying age groups and different socio-cultural milieu.

### 6.2. Limitations

The results of our study have to be interpreted with caution, because it is a cross-sectional study, which does not necessarily allow for drawing causal conclusions. Our results are based on comparing vaccinated against non-vaccinated persons. However, due to the changing situation of the COVID-19 pandemic, these results are time-specific and also depend on the political and social environment of Punjab. In addition, the results may be valid for Punjab, and are not necessarily conferrable to countries with different cultural backgrounds. Lastly, the present study is conducted on a population of individuals aged 40 years and above and might not be generalizable to the broader population.

## Figures and Tables

**Figure 1 ijerph-19-12892-f001:**
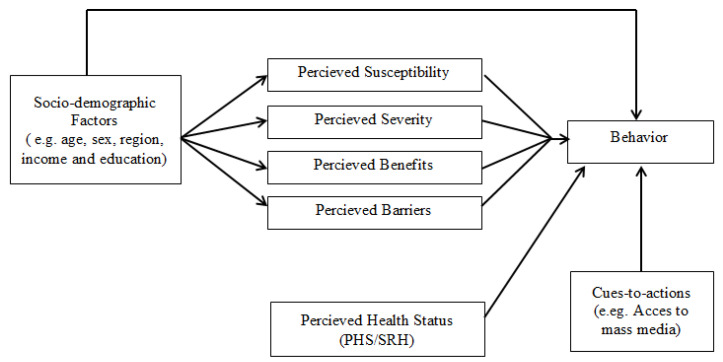
Variables included in the study.

**Table 1 ijerph-19-12892-t001:** Characteristics of study participants.

Characteristics	*n* ^1^	%
Vaccination status		
Vaccinated	795	60.0
Not Vaccinated	530	40.0
Sex		
Male	972	73.4
Female	353	26.6
Age (in years)		
40–49	646	48.8
50–59	421	31.8
60–69	192	14.5
70–79	52	3.9
80+	14	1.1
Mean age (in years) ± SD	51.2 ± 9.34	
Area of residence		
Rural	266	20.2
Urban	1024	77.7
Semi-urban	28	2.1
Religion		
Islam	1270	96.1
Christianity	51	3.9
Others	1	0.1
Monthly household income		
≤PKR 20,000	355	30.3
PKR 20,001–50,000	525	44.7
PKR 50,001–100,000	201	17.1
>PKR 100,000	92	7.8
Marital status		
Currently married	1234	94.1
Not currently married ^2^	78	5.9
Number of children		
0	40	3.2
1–2	289	23.0
3–4	603	48.0
5 and above	325	25.9
Highest level of education		
No formal education	223	16.9
Primary (1–5 years)	109	8.3
Middle (6–10 years)	474	36.0
Secondary (11–12 years)	168	12.8
Higher (13 years and above)	343	26.0
Employment status		
Employed	853	65.1
Not employed ^3^	458	34.9
Access to mass media		
No access	42	3.2
Low access ^4^	235	17.8
Moderate access ^5^	478	36.3
High access ^6^	562	42.7

^1^ Values do not sum up to 1325 in all variables due to missing responses (Refused, Not applicable, I don’t know). ^2^ Includes widow/widower, divorced, separated and single. ^3^ Includes housewife and retired. ^4^ Access to one information source, ^5^ Access to 2–3 information sources. ^6^ Access to 4 or more information sources.

**Table 2 ijerph-19-12892-t002:** Socio-demographic characteristics and HBM.

Characteristics	COVID-19-Related Knowledge				OR (95% CI)	*p*-Value	AOR (95% CI)	*p*-Value
	Poor		Good					
	*n*	%	*n*	%				
Sex								
Male	403	42.2	553	57.8	1		1	
Female	165	57.8	187	42.2	0.82 (0.64–1.05)	0.12	0.68 (0.46–1.01)	0.05
Age (in years)								
40–49	277	43.6	359	56.4	1		1	
50–59	172	41.3	244	58.7	1.09 (0.85–1.40)	0.47	1.10 (0.82–1.46)	0.51
60–69	85	44.5	106	55.5	0.96 (0.69–1.33)	0.81	1.07 (0.72–1.59)	0.71
70–79	25	49.0	26	51.0	0.80 (0.45–1.42)	0.45	1.41 (0.67–2.96)	0.35
80+	9	64.3	5	35.7	0.42 (0.14–1.29)	0.13	0.70 (0.19–2.57)	0.59
Area of residence								
Rural	146	55.5	117	44.5	1		1	
Urban	409	40.4	603	59.6	1.84 (1.39–2.41)	<0.01 *	1.40 (1.02–1.92)	0.03 *
Semi-urban	11	39.3	17	60.7	1.92 (0.87–4.27)	0.10	1.88 (0.74–4.74)	0.17
Monthly household income								
≤PKR 20,000	186	53.0	165	47.0	1		1	
PKR 20,001–50,000	217	41.7	303	58.3	1.57 (1.19–2.06)	<0.01 *	1.32 (0.97–1.77)	0.06
PKR 50,001–100,000	78	39.0	122	61.0	1.76 (1.23–2.51)	<0.01 *	1.08 (0.71–1.64)	0.70
>PKR 100,000	37	40.7	54	59.3	1.64 (1.03–2.62)	0.03 *	0.97 (0.56–1.67)	0.92
Marital status								
Not currently married	46	59.0	32	41.0	1		1	
Currently married	519	42.5	702	57.5	1.94 (1.22–3.09)	<0.01 *	2.03 (1.02–4.03)	0.04 *
Number of children								
0	21	52.5	19	47.5	1		1	
1–2	113	39.5	173	60.5	1.69 (0.87–3.28)	0.12	2.30 (1.04–5.08)	0.03 *
3–4	248	41.5	350	58.5	1.56 (0.82–2.96)	0.17	3.03 (0.94–4.39)	0.06
5 and above	159	49.7	161	50.3	1.11 (0.58–2.16)	0.73	1.62 (0.73–3.57)	0.23
Highest level of education								
No formal education	136	61.8	84	38.2	1		1	
Primary	56	51.4	53	48.6	1.53 (0.96–2.43)	0.07	1.46 (0.88–2.42)	0.14
Middle	190	40.7	277	59.3	2.36 (1.69–3.27)	<0.01 *	1.82 (1.24–2.68)	<0.01 *
Secondary	75	44.9	92	55.1	1.98 (1.32–2.99)	<0.01 *	1.44 (0.87–2.38)	0.15
Higher	107	31.7	231	68.3	3.49 (2.44–4.98)	<0.01 *	2.36 (1.45–3.83)	<0.01 *
Employment status								
Not employed	209	45.7	248	54.3	1		1	
Employed	357	42.6	482	57.4	1.13 (0.90–1.43)	0.27	0.79 (0.54–1.13)	0.20
Access to mass media								
No access	26	61.9	16	38.1	1		1	
Low access	119	51.1	114	48.9	1.55 (0.79–3.05)	0.19	1.27 (0.61–2.64)	0.51
Moderate access	222	47.2	248	52.8	1.81 (0.94–3.47)	0.07	1.22 (0.60–2.49)	0.58
High access	199	35.8	357	64.2	2.91 (1.52–5.56)	<0.01 *	1.59 (0.76–3.32)	0.21

* significant *p*-value.

**Table 3 ijerph-19-12892-t003:** HBM and vaccination status.

Dimensions of Health Belief Model	Total		Vaccinated		Not Vaccinated		OR (95% CI)	*p*-Value
	*n*	%	*n*	%	*n*	%		
P-SUS								
Yes	874	66.1	575	72.6	299	56.4	2.05 (1.62–2.58)	<0.01 *
No	448	33.9	217	27.4	231	43.6	1	
P-SEV								
Yes	930	70.8	586	74.6	344	65.0	1.58 (1.25–2.01)	<0.01 *
No	384	29.2	199	25.4	185	35.0	1	
P-BEN								
Yes	896	67.8	649	81.9	247	46.7	5.18 (4.04–6.65)	<0.01 *
No	425	32.2	143	18.1	282	53.3	1	
P-BAR								
Yes	678	51.4	300	38.0	378	71.3	1	
No	641	48.6	489	62.0	152	28.7	4.05 (3.20–5.14)	<0.01 *
PHS								
Poor	81	6.1	30	3.8	51	9.6	1	
Fair	361	27.3	217	27.3	144	27.2	2.56 (1.56–4.21)	<0.01 *
Good	882	66.6	547	68.9	335	63.2	2.78 (1.73–4.45)	<0.01 *
Access to mass media (cue-to-action)								
No access	42	3.2	21	2.7	21	4.0	1	
Low access	235	17.8	106	13.5	129	24.4	0.82 (0.43–1.59)	0.56
Moderate access	478	36.3	281	35.7	197	37.2	1.43 (0.76–2.68)	0.27
High access	562	42.7	380	48.2	182	34.4	2.09 (1.11–3.92)	0.02 *

* significant *p*-value.

**Table 4 ijerph-19-12892-t004:** Multiple hierarchal regression analysis for factors affecting vaccination uptake.

		Model 1	Model 2	Model 3	Model 4	Model 5	Model 6	Model 7	Model 8
		AOR	AOR	AOR	AOR	AOR	AOR	AOR	AOR
		(95% CI)	(95% CI)	(95% CI)	(95% CI)	(95% CI)	(95% CI)	(95% CI)	(95% CI)
P-SUS	No	(1)							
	Yes	2.07 * (1.64–2.62)	1.97 * (1.56–2.50)	2.20 * (1.70–2.83)	1.47 * (1.11–1.94)	1.49 * (1.13–1.98)	1.68 * (1.23–2.30)	1.63 * (1.19–2.23)	1.54 * (1.11–2.13)
P-SEV	No	(1)							
	Yes		1.43 * (1.12–1.83)	1.50 * (1.15–1.95)	1.29 (0.97–1.70)	1.27 (0.96–1.68)	1.23 (0.90–1.66)	1.26 (0.93–1.72)	1.27 (0.92–1.74)
P-BAR	Yes	(1)							
	No			4.31 * (3.37–5.51)	3.66 * (2.83–4.75)	3.698 * (2.85–4.78)	3.35 * (2.53–4.43)	3.43 * (2.58–4.56)	3.53 * (2.63–4.74)
P-BEN	No	(1)							
	Yes				3.99 * (3.02–5.26)	3.90 * (2.95–5.16)	3.93 * (2.88–5.34)	3.82 * (2.80–5.21)	3.67 * (2.66–5.06)
PHS	Poor	(1)							
	Average					2.82 * (1.57–5.05)	2.88 * (1.48–5.59)	2.95 * (1.50–5.81)	3.05 * (1.51–6.13)
	Good					2.80 *(1.61–4.87)	2.41 *(1.27–4.54)	2.53 *(1.32–4.84)	2.66 *(1.36–5.19)
Age (in years)	40–49	(1)							
	50–59						1.12 (0.82–1.53)	1.20 (0.88–1.65)	1.37 (0.99–1.90)
	60–69						1.73 * (1.13–2.65)	1.84 * (1.19–2.84)	2.13 * (1.36–3.35)
	70–79						0.96 (0.44–2.08)	1.01 (0.46–2.23)	1.06 (0.45–2.48)
	80+						0.79 (0.23–2.75)	0.91 (0.25–3.26)	1.38 (0.37–5.05)
Sex	Male	(1)							
	Female						0.53 * (0.38–0.72)	0.53 * (0.38–0.73)	0.61 * (0.43–0.85)
Region	Rural	(1)							
	Urban						2.34 * (1.66–3.30)	2.17 * (1.53–3.08)	1.74 * (1.20–2.51)
	Semi-urban						0.61 (0.21–1.75)	0.60 (0.21–1.73)	0.45 (0.15–1.34)
Income (in PKR)	≤20,000	(1)							
	20,001–50,000						1.12 (0.80–1.55)	1.02 (0.73–1.43)	0.86 (0.60–1.23)
	50,001–100,000						0.99 (0.65–1.52)	0.81 (0.52–1.26)	0.56 * (0.35–0.91)
	>100,000						0.65 (0.38–1.12)	0.50 * (0.28–0.89)	0.34 * (0.18–0.62)
Access to mass media	No access	(1)							
	Low access							0.57 (0.25–1.28)	0.45 (0.19–1.01)
	Moderate access							1.13 (0.52–2.45)	0.69 (0.31–1.53)
	High access							1.40 (0.64–3.07)	0.71 (0.31–1.60)
Educational level	No formal education	(1)							
	Primary								1.22 (0.67–2.23)
	Middle								3.92 * (2.50–6.16)
	Secondary								4.38 * (2.44–7.84)
	Higher								4.53 * (2.60–7.90)

* significant *p*-value.

## Data Availability

The data presented in this study are available upon reasonable request from the corresponding author.

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
