# Peer review of "COVID-19 Vaccination Acceptance in the Context of the Health Belief Model: Comparative Cross-Sectional Study in Punjab, Pakistan"

_ijerph, 2022, doi:10.3390/ijerph191912892_

Round 1
Reviewer 1 Report
Dear authors, it is an honor to have the opportunity to comment on your work. The COVID-19 pandemic has caused a lot of inconvenience to our lives, and vaccines can effectively reduce the threat we face from the virus. So I agree that research into people's willingness to vaccinate is urgent and important. My comments on this article mainly focus on two parts: survey method and writing structure.
First of all, I am most confused about the data survey and statistical methods, as shown below.
1. The precondition of multiple regression analysis is that the data type of the dependent variable is continuous variable, while the dependent variable in this paper seems to be a categorical variable (Vaccination status). Why not use logistic regression or Stata Multinomial Logit/Probit regression for statistical analysis? These seem to be appropriate statistical methods when the dependent variable is categorical.
2. Is there any literature basis for the questionnaire? SRH and PHS do not mention specific items in the manuscript. On the one hand, p-sus, p-sev, p-ben and p-bar seem to adopt a more subjective way of asking questions. Absence of reference may cause the question to deviate from your definition of construct.
3. Why should SRH and PHS be investigated with 5-point scale and then merged into three categories instead of directly asking questions with 3-point scale?
4. When subjects were unsure about p-sev were they placed in the YES category or the NO category?
5. When subjects answered one yes and one no to p-ben's two questions, should he be placed in the YES or No category?
6. The manuscript mentions that p-bar has two items for measurement, but only one item is reported.
Second, there are comments on the writing style and structure, as follows.
7. I think a literature review section should be added to introduce the research variables and theoretical models involved in this paper in detail. All variables that appear in the manuscript should be clearly defined.
8. A figure can be added to the section Materials and Methods to illustrate the variables involved in this paper. This will help readers to understand the content of the study more clearly.
9. I suggest the author give a more detailed explanation of the definition and calculation formula of Odds and Adjusted Odd Ratios. These two values seem to be the core of the data calculation in this paper. A clear explanation of statistical methods will help readers unfamiliar with quantitative calculations understand this article.
10. I propose to supplement the two new sections on management contribution and theoretical contribution. You can further explain the need to publish your work in these two sections.
11. Please add two sections in Conclusion: future research and research limitations.
Author Response
Thank you very much for your recommendations. Please find the detailed response attached.

Reviewer 2 Report
The authors reported their latest research article titled ‘COVID-19 Vaccination Acceptance in the Context of the Health Belief Model: Comparative Cross-Sectional Study in Punjab, Pakistan’ for publication in the International Journal of Environmental Research and Public Health. They assessed the role of Health Belief Model constructs and perceived health status on the vaccination status of individuals and to understand the role of socio-18 demographic variables on Health Belief Model scoring. This manuscript provides fascinating insight along with comprehensive examinations. On the other hand, some comments need to be addressed before publication.
1. More explanation should be given as to why the study was conducted among individuals aged 40 and over. In addition, conducting studies in this age range makes it difficult to evaluate the general situation. Because when it comes to the younger population, many parameters will change. Therefore, the age range studied should be emphasized in the content and title.
2. In the text, the survey consisted of 6 sections, but only 4 of the relevant sections were suitable for this analysis. If 4 sections are related to the study, why was the survey created in 6 sections?
3. In the text, there is a sentence that the role of access to mass media was evaluated as insignificant as a result of the study. On the contrary, as a result of the study, one of the criteria for getting a good score was accepted as high media access. These statements contradict each other.
4. The same sentences are repeated too many times in the text. It needs careful revision in the whole manuscript.
Author Response

(The authors gave the same response as above.)

Reviewer 3 Report
Dear authors,
The topic of your paper and the way you presented your research is an interesting one and congratulations are in order, however, there are some issues of your paper that should be corrected:
- From my point of view the paper should also have a brief literature review section where it covers in a more extensive way the Health Belief Model, with its components, implications etc.
- Secondly the discussion section, from my perspective should not include so many conclusions. The conclusion section should be more extensive (could include some of the elements of the discussion). Also in the conclusion section one should also present the implications of the study and one should also present aspects related to how to benefit from its results. Your paper is very interesting one and the study results could be beneficial for the public health professionals.
Author Response

(The authors gave the same response as above.)

Reviewer 4 Report
Dear Authors,
I find the work relevant and sound scientifically speaking.
I have only a few minor comments:
1. In the introduction part COVID-19 vaccine hesitancy could be addressed, because the vaccine refusal is actually an expression of it and there is a large literature available related to HBM.
2. What is the objective or the hypotheses of the study? I think they need to be highlighted more clearly.
3. The HBM factors as predictors of COVID-19 vaccination uptake were addressed in some studies that I do not see in the citations and I believe it can add a few ideas to the closing part. The following articles are relevant:
Limbu, Y. B., Gautam, R. K., & Pham, L. (2022). The Health Belief Model Applied to COVID-19 Vaccine Hesitancy: A Systematic Review. Vaccines, 10(6), 973. https://doi.org/10.3390/vaccines10060973
Chen H, Li X, Gao J, Liu X, Mao Y, Wang R, Zheng P, Xiao Q, Jia Y, Fu H, Dai J
Health Belief Model Perspective on the Control of COVID-19 Vaccine Hesitancy and the Promotion of Vaccination in China: Web-Based Cross-sectional Study J Med Internet Res 2021;23(9):e29329
Marschalko EE, Szabo K, Kotta I and Kalcza-Janosi K (2022) The Role of Positive and Negative Information Processing in COVID-19 Vaccine Uptake in Women of Generation X, Y, and Z: The Power of Good is Stronger Than Bad in Youngsters? Front. Psychol. 13:925675. doi: 10.3389/fpsyg.2022.925675
2. All tables should be presented along with explanation notes on factors for a clearer interpretation.
4. An image/figure which includes the most important predictors could help the reader's experience.
5. In some parts of the article it is used the "effect" word linked to Regression Modelling and can be linked to causality relationship. The "effects" around HBM need to be changed to "role of" or "influence of". The design is correlational and needs to be presented as such.
Thank you!
Author Response

(The authors gave the same response as above.)

Reviewer 5 Report
In this study, the authors have conducted a survey regarding COVID-19 vaccination acceptance in the Punjab, Pakistan. The results of this study will be useful for scientists working on preventive measures against COVID-19 and other infectious diseases. The article is suitable for publication.
This article uses Health belief model to analyse vaccine uptake. It has the aim to find out the role of HBM constructs on vaccination status and also the role played by socio-demographic factors on HBM constructs. The topic is relevant in the present context of large scale covid vaccination carried out in many countries. Findings of the study will help to modify measures to promote vaccine adoption. Higher educational level was found to be the strongest predictor of vaccine uptake. The authors have presented data in proper tables and carried out appropriate statistical analysis. The conclusions made in the article reflect the data generated in the study.Corrections suggested below may be carried out before final submission.
Corrections/modifications required
Section |
Line No. |
Corrections |
Discussion |
235 |
which have beeb to be corrected as which have been |
|
307 |
designed in a way that target the. – sentence not complete |
Note: Corrected/added words are underlined.
Author Response

(The authors gave the same response as above.)

Reviewer 6 Report
In addition, this study can be greatly strengthened by addressing following points:
1.Line 24 and Line 301, Please write the full name of ICT for the first time.
2.Line 204, ” P-BEN (AOR=3.35, 95% CI: 2.53–4.43)”, the P-BEN should AOR=3.93, 95% CI:2.88-5.34 in the table 4?
3.In my opinion, gender and age are important factors affecting vaccination. However, in this study the sex ratio in this study is 1: 2, why?
4.The impact of gender should be also discussed in Line 203-209.
Author Response

(The authors gave the same response as above.)

Round 2
Reviewer 1 Report
The authors addressed my comments. I recommend publishing this article as it is a good contribution to the topic and well structured.
Congratulations.